# A Conway–Maxwell–Poisson-Binomial AR(1) Model for Bounded Time Series Data

**DOI:** 10.3390/e25010126

**Published:** 2023-01-07

**Authors:** Huaping Chen, Jiayue Zhang, Xiufang Liu

**Affiliations:** 1School of Mathematics and Statistics, Henan University, Kaifeng 475004, China; 2School of Mathematics, Jilin University, Changchun 130012, China; 3College of Mathematics, Taiyuan University of Technology, Taiyuan 030024, China

**Keywords:** CMPB thinning operator, bounded time series, CMPBAR model, under-dispersion, equi-dispersion, over-dispersion

## Abstract

Binomial autoregressive models are frequently used for modeling bounded time series counts. However, they are not well developed for more complex bounded time series counts of the occurrence of *n* exchangeable and dependent units, which are becoming increasingly common in practice. To fill this gap, this paper first constructs an exchangeable Conway–Maxwell–Poisson-binomial (CMPB) thinning operator and then establishes the Conway–Maxwell–Poisson-binomial AR (CMPBAR) model. We establish its stationarity and ergodicity, discuss the conditional maximum likelihood (CML) estimate of the model’s parameters, and establish the asymptotic normality of the CML estimator. In a simulation study, the boxplots illustrate that the CML estimator is consistent and the qqplots show the asymptotic normality of the CML estimator. In the real data example, our model takes a smaller AIC and BIC than its main competitors.

## 1. Introduction

Bounded time series of counts are commonly observed in real-world applications. Its (binomial) index of dispersion (as a function of *n*, μ and $\sigma^{2}$) is defined by $\mathrm{BID}\left(X\right)=n\sigma^{2}/\left(\mu (n-\mu )\right),$ where *n* is the predetermined upper limit of the range, E(X)=μ and $\mathrm{Var}\left(X\right)=\sigma^{2}$. If its BID(X)<1, then it is under-dispersed, if its BID(X)=1, then it is equi-dispersed, while if its BID(X)>1, then it is over-dispersed (or the extra-binomial variation).

A popular tool to establish a binomial autoregressive model (BAR) is the binomial thinning operator “∘” [1], which is introduced by
(1)$$\begin{array}{c} \alpha \circ X:=\sum_{i=1}^{X}W_{i}, \end{array}$$
where *X* is a non-negative integer-valued random variable, $\left\{W_{i},i=1,2,\cdots ,n\right\}$ is an i.i.d. Bernoulli random variable sequence with $P\left(W_{i}=1\right)=1-P\left(W_{i}=0\right)=\alpha$ and independent of *X*. McKenzie [2] used the binomial thinning operator given in (Equation 1) to establish the binomial AR(1) model, which is a popular model for bounded time series and defined as follows
(2)$$\begin{array}{c} X_{t}=\alpha \circ X_{t-1}+\beta \circ \left(n-X_{t-1}\right), \end{array}$$
where n∈N is the predetermined upper limit of the range; $X_{0}$ follows the binomial distribution with P(X0=k)=nkπk(1−π)n−k; α=β+ρ and β=(1−ρ)π with $\rho \in \left(\max\left\{-\pi /(1-\pi ),-(1-\pi )/\pi \right\},1\right)$ and π∈(0,1); the counting series at time *t* are independent of the random variables $X_{s},\forall s<t$; and all the counting series in “α∘” and “β∘” are mutually independent sequences of independent Bernoulli random variables with parameters α and β, respectively. The binomial AR(1) process given in (Equation 2) is now well understood and it is an ergodic Markov chain with a stationary distribution Bin(n,π) with π=β/(1−ρ) and ρ=α−β. Hence, its $\mathrm{BID}(X_{t})=1$, i.e., the BAR model given in (Equation 2), applies to equi-dispersed time series with finite range; see [3,4,5,6,7] for more discussion about the BAR(1) model.

Weiß and Pollett [8] extended the binomial AR(1) model as the density-dependent BAR(1) model (denoted as the DDBAR(1) model), whose thinning probabilities vary over time by assuming $\alpha_{t}=\alpha \left(X_{t-1}/n\right)$ and $\beta_{t}=\beta \left(X_{t-1}/n\right)$. In particular, for given *n*, if $\alpha_{t}=\left(1-\rho \right)\left(a+bX_{t-1}/n\right)$ and $\beta_{t}=\left(1-\rho \right)\left(a+bX_{t-1}/n\right)+\rho$, the DDBAR(1) model allows to analyze bounded integer-valued time series with under-dispersion, equi-dispersion and over-dispersion, see Section 4 in [8] for more details. To model extra-binomial variation for time series counts, Weiß and Kim [9] proposed the beta-binomial AR (BBAR) model based on the beta-binomial thinning operator “$\alpha \circ_{\phi }$”, which is introduced by
$$\begin{array}{c} \alpha_{\phi }\circ X=\sum_{i=1}^{X}B_{i}, \end{array}$$
where *X* is a non-negative integer-valued random variable, $\left\{B_{i},i=1,2,\cdots ,n\right\}$ is an i.i.d. Bernoulli random variable sequence with $P\left(B_{i}=1|\alpha_{\phi }\right)=1-P\left(B_{i}=0|\alpha_{\phi }\right)=\alpha_{\phi }$ and $\alpha_{\phi }∼\mathrm{Beta}\left(\tau \alpha ,\tau \left(1-\alpha \right)\right),\;\tau =\left(1-\phi \right)/\phi$, $\left\{B_{i},i=1,2,\cdots ,X\right\}$ is independent of *X*.

As discussed in Weiß [10], the BAR(1) model, DDBAR(1) model, and BBAR(1) model can be interpreted as a system with *n* mutually independent units and each unit being either in state “1” or state “0”. Assume $X_{t}$ is the number of units being in state “1” at time *t*. Then $\alpha \circ X_{t-1}$ ($\alpha_{t}\circ X_{t-1}$ or $\alpha_{\phi }\circ X_{t-1}$) is the number of units still in state “1” at time *t* with survival probability α (random survival probability $\alpha_{t}$ or $\alpha_{\phi }$), $\beta \circ (n-X_{t-1})$ ($\beta_{t}\circ \left(n-X_{t-1}\right)$ or $\beta_{\phi }\circ \left(n-X_{t-1}\right)$) is the number of units, which moved from state “0” to state “1” at time *t* with revival probability β (random revival probability $\beta_{t}$ or $\beta_{\phi }$). It is worth mentioning that all of BAR(1), DDBAR(1), and BBAR(1) models are aimed at a system with *n* independent units, but not a system with *n* dependent units, i.e., the counting series in “∘” is independent and identically distributed, but not dependent. To solve this dilemma, Kang et al. [11] proposed a generalized binomial AR (GBAR) model based on the generalized binomial thinning operator “$\alpha \circ_{\theta }$”, which is proposed by Ristić et al. [12] and given as follows
$$\begin{array}{c} \alpha_{\theta }\circ X=\sum_{i=1}^{X}U_{i}, \end{array}$$
where $U_{i}=\left(1-V_{i}\right)W_{i}+V_{i}Z$, $\left\{W_{i}\right\}$ and $\left\{V_{i}\right\}$ are two independent random sequences of iid random variables with Bernoulli(α) and Bernoulli(θ) distributions, *Z* is a Bernoulli(α) random variable and is responsible for the cross-dependence, ∀i,j=1,2,...,X, $\left\{W_{i}\right\}$, $\left\{V_{j}\right\}$ and *Z* are mutually independent and each of them is independent of *X*.

Unfortunately, the GBAR model [11] can not use to analyze under-dispersed or equi-dispersed bounded data. To fill this gap, we are inspired by the Conway–Maxwell–Poisson-binomial (CMPB) distribution [13] and construct the Conway–Maxwell–Poisson-binomial thinning operator, whose counting series is exchangeablility. Furthermore, we propose a new Conway–Maxwell–Poisson-binomial autoregressive (CMPBAR) model, which not only allows us to analyze bounded data with over-dispersion but also allows us to model bounded data with equi-dispersion or under-dispersion. The second contribution of this paper is that we discuss the CML estimation of the parameters involved in the new model, and establish the asymptotic normality of the CML estimator. To illustrate that the new model is more flexible and superior, we apply the new model on the weekly rainy days at Hamburg–Neuwiedenthal in Germany.

The paper is organized as follows. Section 2 first gives a brief review of the Conway–Maxwell–Poisson-binomial distribution, then gives the definition of the exchangeable Conway–Maxwell–Poisson-binomial thinning operator and that of the Conway–Maxwell–Poisson-binomial AR model. The conditional maximum likelihood estimation and its asymptotic properties are established in Section 3. Section 4 gives a simulation study and Section 5 gives real data to show the better performance of the new model. Conclusions are made in Section 6.

## 2. Model Formulation and Stability Properties

### 2.1. Conway–Maxwell–Poisson-Binomial Distribution

For readability, we first give a brief review of the CMPB distribution introduced by Shmueli et al. [13].

A random variable *X* taking values in {0,1,2,…,n} is said to follow the Conway–Maxwell–Poisson-binomial distribution with parameters (α,ν), if the probability mass function (pmf) of *X* takes the form P(X=x|α,ν,n)=nxναx(1−α)n−x/Z(α,ν), where Z(α,ν)=∑x=0nnxναx(1−α)n−x, 0<α<1, ν∈R and n∈N is the predetermined upper limit of the range.

For simplicity, we write X∼CMPB(n,α,ν). Denote θ=α/(1−α), the pmf of *X* can be rewritten as
(3)P(X=x|θ,ν,n)=1S(θ,ν)nxνθx,
where S(θ,ν)=∑x=0nnxνθx, θ>0 and n∈N is the predetermined upper limit of the range. Therefore, we obtain the moment-generating function of *X* as $M_{X}\left(s\right)=E\left(e^{sX}\right)=\frac{S(\theta e^{s},\nu )}{S(\theta ,\nu )}$. Furthermore,
(4)$$\begin{array}{cc} \; & E\left(X\right)=\theta \frac{S^{\prime }\left(\theta ,\nu \right)}{S(\theta ,\nu )},\;\mathrm{Var}\left(X\right)=\theta \frac{S^{\prime }\left(\theta ,\nu \right)}{S(\theta ,\nu )}+\theta^{2}\left(\frac{S^{\prime \prime }\left(\theta ,\nu \right)}{S(\theta ,\nu )}-{\left(\frac{S^{\prime }\left(\theta ,\nu \right)}{S(\theta ,\nu )}\right)}^{2}\right),\\ \; & \mathrm{BID}=\frac{n\mathrm{Var}(X)}{E\left(X\right)\left(n-E(X)\right)}=\frac{S\left(\theta ,\nu \right)S^{\prime }\left(\theta ,\nu \right)+\theta S\left(\theta ,\nu \right)S^{\prime \prime }\left(\theta ,\nu \right)-\theta {\left(S^{\prime }\left(\theta ,\nu \right)\right)}^{2}}{nS\left(\theta ,\nu \right)S^{\prime }\left(\theta ,\nu \right)-\theta {\left(S^{\prime }\left(\theta ,\nu \right)\right)}^{2}}, \end{array}$$
where $S^{\prime }\left(\theta ,\nu \right)=\partial S\left(\theta ,\nu \right)/\partial \theta$ and $S^{\prime \prime }\left(\theta ,\nu \right)=\partial S^{\prime }\left(\theta ,\nu \right)/\partial \theta$ (see Shmueli et al. [13], Borges et al. [14], Daly and Gaunt [15], and Kadane [16] for more detailed discussion).

Unfortunately, the specific range of the BID for the CMPB distribution can not be obtained by (Equation 4). To solve this dilemma, we give an example in Figure 1 with n=7, when α and ν are varying from {0.1,0.2,0.3,⋯,0.9} and {−2,−1.5,−0.5,0,0.5,1,1.5,2,2.5}, respectively.

From Figure 1, the BID of the CMPB distribution takes a value, which may be less than 1, equal to 1, or greater than 1 for different values α and ν. Additionally, it implies that the CMPB distribution allows us to analyze bounded time series counts with under-dispersion, equi-dispersion, and over-dispersion.

To further explore the dynamic change of the BID with α varying from {0.1,0.2,⋯,0.9} for given n=7 and ν=−0.5, 0, 0.5, 1, 1.5, or 2, we present the plots of the BID in Figure 2.

From Figure 2, we obtain the following observations. First, if ν<1, the BID is no less than 1. To be precise, its BID is increasing to maximum when α is varying from 0 to 0.5, and then decreasing to 1 when α is varying from 0.5 to 1. Second, if ν=1, its BID = 1, for all α∈(0,1). Third, if ν>1, its BID is no more than 1. Precisely, its BID is decreasing to the minimum when α is varying from 0 to 0.5, and then increasing to 1 when α is varying from 0.5 to 1. To sum up, the Conway–Maxwell–Poisson-binomial distribution allows under-dispersion, equi-dispersion, and over-dispersion for bounded time series data.

**Remark** **1.**
*By (Equation 3), the pmf of the CMPB (n,α,ν) is expressed as that of the power series distribution and if ν=0, $P\left(X=x|\theta ,\nu ,n\right)=\theta^{x}/\sum_{x=0}^{n}\theta^{x},\theta =\alpha /\left(1-\alpha \right)$, if ν=1, the CMPB(n,α,ν) reduces to binomial distribution with parameter α.*


### 2.2. Conway–Maxwell–Poisson-Binomial Thinning Operator

By Shmueli et al. [13], the CMPB distribution is a distribution on the sum of *n* dependent Bernoulli components without specifying anything else about the joint distribution of those components. Precisely, if X∼CMPB(n,α,ν), there exists a Bernoulli variable sequence $\left\{Z_{i}\right\}$ such that $X=\sum_{i=1}^{n}Z_{i}$, where
(5)Pz1,⋯,zn:=P(Z1=z1,⋯,Zn=zn)=1∑z1=01⋯∑zn=01nxν−1θxnxν−1θx
with $\theta =\alpha /\left(1-\alpha \right),\;x=\sum_{i=1}^{n}z_{i}$ and $\left(z_{1},z_{2},\cdots ,z_{n}\right)\in {\left\{0,1\right\}}^{n}$.

**Definition** **1.**
*Let θ=α/(1−α). Then the exchangeable Conway–Maxwell–Poisson-binomial thinning operator is introduced by*

(6)
$$\begin{array}{c} \alpha {⋄}_{\nu }X:=\sum_{i=1}^{X}Z_{i}, \end{array}$$

*where X is a non-negative random variable, $\left\{Z_{i},\;i=1,2,\cdots ,X\right\}$ is an exchangeable Bernoulli variable sequence with its pmf taking the form (Equation 5) and independent of X.*


To generate the random number of “$\alpha {⋄}_{\nu }X$”, we first let X=n, then $\alpha {⋄}_{\nu }X|\left(X=n\right)∼\mathrm{CMPB}\left(n,\alpha ,\nu \right)$. Therefore, $E\left(\alpha {⋄}_{\nu }X|X=n\right)=\theta S^{\prime }\left(\theta ,\nu \right)/S(\theta ,\nu ),$ $\mathrm{Var}\left(\alpha {⋄}_{\nu }X|X=n\right)=\theta \frac{S^{\prime }\left(\theta ,\nu \right)}{S(\theta ,\nu )}+\theta^{2}\left(\frac{S^{\prime \prime }\left(\theta ,\nu \right)}{S(\theta ,\nu )}-{\left(\frac{S^{\prime }\left(\theta ,\nu \right)}{S(\theta ,\nu )}\right)}^{2}\right)$ and the conditional binomial index of dispersion (CBID) is $\mathrm{CBID}=\frac{S\left(\theta ,\nu \right)S^{\prime }\left(\theta ,\nu \right)+\theta S\left(\theta ,\nu \right)S^{\prime \prime }\left(\theta ,\nu \right)-\theta {\left(S^{\prime }\left(\theta ,\nu \right)\right)}^{2}}{nS\left(\theta ,\nu \right)S^{\prime }\left(\theta ,\nu \right)-\theta {\left(S^{\prime }\left(\theta ,\nu \right)\right)}^{2}},$ where S(θ,ν)=∑x=0nnxνθx, $S^{\prime }\left(\theta ,\nu \right)=\partial S\left(\theta ,\nu \right)/\partial \theta$, and $S^{\prime \prime }\left(\theta ,\nu \right)=\partial S^{\prime }\left(\theta ,\nu \right)/\partial \theta$.

Second, we let θ=α/(1−α), then the pmf of $\alpha {⋄}_{\nu }n$ takes the form (Equation 3). Third, we let $\theta =\lambda^{\nu }$, λ>0. By (Equation 3), the pmf of the $\alpha {⋄}_{\nu }n$ can be rewritten as
P(α⋄νn=x)=1U(λ,ν)nxλxνwithU(λ,ν)=∑x=0nnxλxν.
Furthermore,
(7)$$\begin{array}{c} P\left(\alpha {⋄}_{\nu }n=x+1\right)={\left(\frac{n-x}{x+1}\lambda \right)}^{\nu }P\left(\alpha {⋄}_{\nu }n=x\right), \end{array}$$
by which an algorithm is used to generate a random number of $\alpha {⋄}_{\nu }X$ with X=n can be expressed as follows.

**Remark** **2.**
*By Kadane [16], the counting series $\left\{Z_{i}\right\}$ in Definition 1 is a dependent Bernoulli variable sequence with exchangeability of order 2. To account for the concept of exchangeability, we assume π is a permutation of $\left(z_{1},z_{2},\cdots ,z_{n}\right)$. Then $P_{z_{1},\cdots ,z_{n}}=P_{\pi {(}_{1},\cdots ,z_{n})}$. By the definition of exchangeability in Section 6 in Kadane [16], $\sum_{i=1}^{n}Z_{i}$ is n-exchangeable. Kadane [16] stated that “de Finetti’s Theorem shows that sums of exchangeable random variables are mixtures of Binomial random variables. Because the marginal distribution of each component is Bernoulli, interest centers on the joint distribution of pairs of such variables”. By Theorem 4 in Kadane [16], n-exchangeability applies to every permutation of length n, it implies that $n^{\prime }$ is exchangeable for each $n^{\prime }<n$. Hence, $\left\{Z_{i}\right\}$ is exchangeable with order 2 because every pair has the same distribution as every other pair, i.e., every pair of $\left\{Z_{1},Z_{2},\cdots ,Z_{n}\right\}$ has the same distribution as every other pair and for any pair $\left(Z_{i},Z_{j}\right)$, ∀i,j=1,2,⋯,n, and i≠j, $P\left(Z_{i}=0,Z_{j}=1\right)=P\left(Z_{i}=1,Z_{j}=0\right)>0$, $P\left(Z_{i}=0,Z_{j}=0\right)+2P\left(Z_{i}=0,Z_{j}=1\right)+P\left(Z_{i}=1,Z_{j}=1\right)=1$, $P(Z_{i}=1,Z_{j}=1)>0$, and $P(Z_{i}=0,Z_{j}=0)>0$; see [16] for more discussion.*


### 2.3. Binomial Autoregressive Model with the CMPB Operator

Now, we define the BAR(1) model with the CMPB operator by
(8)$$\begin{array}{c} X_{t}=\alpha {⋄}_{\nu }X_{t-1}+\beta {⋄}_{\nu }\left(n-X_{t-1}\right), \end{array}$$
where 0<α<1, 0<β<1, both $\alpha {⋄}_{\nu }X_{t-1}=\sum_{i=1}^{X_{t-1}}Z_{i}$ and $\beta {⋄}_{\nu }\left(n-X_{t-1}\right)=\sum_{i=1}^{n-X_{t-1}}W_{i}$ are the CMPB thinning operators given in Definition 1, their counting series $\left\{Z_{i}\right\}$ and $\left\{W_{i}\right\}$ are the exchangeable Bernoulli variable sequence with their pmfs taking the form (Equation 5), $\left\{Z_{i}\right\}$ is independent of $\left\{W_{j}\right\}$, $\forall i=1,2,\ldots ,X_{t-1}$, $j=1,2,\ldots ,(n-X_{t-1})$, and all the thinnings at time *t* are independent of $\left\{X_{s},s<t\right\}$, n∈N, ν∈R.

For simplicity, we denote the new model as the CMPBAR(1) model. By (Equation 8), ${\left\{X_{t}\right\}}_{N}$ is a Markov chain and its one-step transition probability takes the form
(9)Pη(k|l)=P(Xt=k|Xt−1=l)=1S(θ1,ν)S(θ2,ν)∑i=0min{k,l}liνn−lk−iνθ1iθ2k−i,
where S(θ1,ν)=∑i=0lliνθ1i and S(θ2,ν)=∑i=0n−ln−liνθ2i with $\eta =(\theta_{1},\theta_{2},\nu )$ and $\theta_{1}=\alpha /\left(1-\alpha \right)$ and $\theta_{2}=\beta /\left(1-\beta \right)$.

**Theorem** **1.**
*If $\left\{X_{t}\right\}$ satisfies (Equation 8), then $\left\{X_{t}\right\}$ is ergodicity and strictly stationarity.*


**Proof.** Similar to that of Theorem 1 in Kang et al. [11], the state space of $\left\{X_{t}\right\}$ is {0,1,⋯,n}. Because $P\left(X_{t}=i|X_{t-1}=j\right)>0,\forall i,j\in \left\{0,1,\cdots ,n\right\}$, so the state space of $\left\{X_{t}\right\}$ is an equivalence class. Furthermore, $\left\{X_{t}\right\}$ is an irreducible and aperiodic Markov chain; therefore, $\left\{X_{t}\right\}$ is ergodic with a unique stationary distribution by [17]. □

By Definition 1 and (Equation 8), for given $X_{t-1}$, $\left\{X_{t}\right\}$ given in (Equation 8) consists of two independent parts $\alpha {⋄}_{\nu }X_{t-1}$ and $\beta {⋄}_{\nu }\left(n-X_{t-1}\right)$, where $\alpha {⋄}_{\nu }X_{t-1}∼\mathrm{CMPB}\left(X_{t-1},\alpha ,\nu \right)$ and $\beta {⋄}_{\nu }\left(n-X_{t-1}\right)∼\mathrm{CMPB}\left(n-X_{t-1},\beta ,\nu \right)$. Denote $\theta_{1}=\alpha /\left(1-\alpha \right)$ and $\theta_{2}=\beta /\left(1-\beta \right)$. Then
$$\begin{array}{cc} \; & E\left(X_{t}|X_{t-1}\right)=\theta_{1}S_{1}^{{}^{\prime }}/S_{1}+\theta_{2}S_{2}^{{}^{\prime }}/S_{2},\\ \; & \mathrm{Var}\left(X_{t}|X_{t-1}\right)=\theta_{1}\frac{S_{1}^{{}^{\prime }}}{S_{1}}+\theta_{2}\frac{S_{2}^{\prime }}{S_{2}}+\theta_{1}^{2}\left(\frac{S_{1}^{\prime \prime }}{S_{1}}-{\left(\frac{S_{1}^{\prime }}{S_{1}}\right)}^{2}\right)+\theta_{2}^{2}\left(\frac{S_{2}^{\prime \prime }}{S_{2}}-{\left(\frac{S_{2}^{\prime }}{S_{2}}\right)}^{2}\right), \end{array}$$
and the conditional binomial index of dispersion (CBID) is
$$\begin{array}{c} \mathrm{CBID}=\frac{\theta_{1}^{2}S_{2}^{2}\left(S_{1}S_{1}^{\prime \prime }-{\left(S_{1}^{\prime }\right)}^{2}\right)+\theta_{2}^{2}S_{1}^{2}\left(S_{2}S_{2}^{\prime \prime }-{\left(S_{2}^{\prime }\right)}^{2}\right)+\theta_{1}S_{1}S_{1}^{\prime }S_{2}^{2}+\theta_{2}S_{2}S_{2}^{\prime }S_{1}^{2}}{\left(nS_{1}S_{2}-\theta_{1}S_{1}^{\prime }S_{2}-\theta_{2}S_{1}S_{2}^{\prime }\right)\left(\theta_{1}S_{2}S_{1}^{\prime }+\theta_{2}S_{1}S_{2}^{\prime }\right)} \end{array}$$
where S1:=S1(θ1,ν)=∑x=0Xt−1Xt−1xνθ1x, $S_{1}^{\prime }:=S_{1}^{\prime }\left(\theta_{1},\nu \right)=\partial S_{1}\left(\theta_{1},\nu \right)/\partial \theta_{1}$, $S_{1}^{\prime \prime }:=S_{1}^{\prime \prime }\left(\theta_{1},\nu \right)=\partial S_{1}^{\prime }\left(\theta_{1},\nu \right)/\partial \theta_{1}$, S2:=S2(θ2,ν)=∑x=0n−Xt−1n−Xt−1xνθ2x, $S_{2}^{\prime }:=S_{2}^{\prime }\left(\theta_{2},\nu \right)=\partial S_{2}\left(\theta_{2},\nu \right)/\partial \theta_{2}$, $S_{2}^{\prime \prime }:=S_{2}^{\prime \prime }\left(\theta_{2},\nu \right)=\partial S_{2}^{\prime }\left(\theta_{2},\nu \right)/\partial \theta_{2}$.

Unfortunally, because of the complexity of $S_{1}\left(\theta_{1},\nu \right)$ and $S_{2}\left(\theta_{2},\nu \right)$, we can not obtain the marginal distribution of $\left\{X_{t}\right\}$ and its the autocorrelation structure, including the $E(X_{t})$, $\mathrm{Var}(X_{t})$, and BID. To resolve this dilemma, for given n=10, we create some plots of the BID (in Figure 3) by generating some samples from the CMPBAR(1) model with ν∈{−5,−4.5,−4,⋯,4.5,5} and sample size T=500, when (α,β) = (0.2, 0.2), (0.2, 0.5), (0.2, 0.6), (0.5, 0.6), i.e., $\left(\theta_{1},\theta_{2}\right)$ = (0.25, 0.25), (0.25, 1), (0.25, 1.5), (1, 1.5).

From Figure 3, we have the following observations. First, if ν<1, the BID of the CMPBAR(1) model is greater than 1, i.e., the CMPBAR(1) model allows us to analyze bounded integer-valued time series with overdispersion. Second, if ν>1, the BID of the CMPBAR(1) model is less than 1, i.e., the CMPBAR(1) model allows us to analyze bounded integer-valued time series with underdispersion. Third, if ν=1, the CMPBAR(1) model becomes to the BAR(1) given in (Equation 2) and its BID is equal to 1, i.e., equi-dispersed bounded integer-valued time series is allowed.

## 3. Parameter Estimation

In this section, we use the conditional maximum likelihood method to estimate the parameters (denoted as $\eta ={\left(\theta_{1},\theta_{2},\nu \right)}^{⊤}$) involving in the CMPBAR(1) model. Let $\left\{X_{0},X_{1},\ldots ,X_{T}\right\}$ be a realization of $\left\{X_{t}\right\}$, and generate by the CMPBAR(1) process based on Algorithm 1, where T∈N represents the size of sample.

**Algorithm 1:** Random number generation algorithm for the CMPB distribution
Step 1.generate a random number *u*, u∼Uniform(0,1);Step 2.$x=0,p=P(\alpha {⋄}_{\nu }n=0|\theta ,\nu ,n)$, F=p, where $P(\alpha {⋄}_{\nu }n=0|\theta ,\nu ,n)$ is given in (Equation 3);Step 3.if u<F, set $\alpha {⋄}_{\nu }n=x$ and stop;Step 4.else $p=p\times {\left(\frac{n-x}{x+1}\lambda \right)}^{\nu }$ by (Equation 7), F=F+p,x=x+1;Step 5.go to Step 3.


By using (Equation 9), the conditional log-likelihood function can be written as:(10)ℓ(η)=∑t=1TlogPη(Xt|Xt−1)=∑t=1Tlog∑i=0mXt−1iνn−Xt−1Xt−iνθ1iθ2Xt−i−log(S(θ1,ν))−log(S(θ2,ν)),
where S(θ1,ν)=∑i=0Xt−1iXt−1νθ1i and S(θ2,ν)=∑i=0n−Xt−1in−Xt−1νθ2i with *m* = $\min\left\{X_{t},X_{t-1}\right\}$, $\theta_{1}>0$, $\theta_{2}>0$, and ν∈R. Then the CML estimate ${\hat{\eta }}^{cml}$ is obtained by minimizing (Equation 10).

**Assumption** **1.**
*If there exists a t≥1, such that $X_{t}\left(\eta \right)=X_{t}\left(\eta_{0}\right)$, $P_{\eta_{0}}$ a.s., then $\eta =\eta_{0}$, where $P_{\eta_{0}}$ is the probability measure under the true parameter $\eta_{0}$ with $\eta_{0}=\left\{\theta_{1}^{0},\theta_{2}^{0},\nu^{0}\right\}$.*


**Theorem** **2.**
*Let $\left\{X_{t}\right\}$ be generalized by the CMPBAR(1) model. If Assumption 1 holds, there exists an estimator ${\hat{\eta }}^{cml}$ such that*

$${\hat{\eta }}^{cml}\;\overset{a.s.}{\to }\;\eta_{0}\;and\;\sqrt{T}\left({\hat{\eta }}^{cml}-\eta_{0}\right)\overset{d}{⟶}N\left(0,J^{-1}\left(\eta_{0}\right)I\left(\eta_{0}\right)J^{-1}\left(\eta_{0}\right)\right),T\to \infty ,$$


*where $I\left(\eta_{0}\right)=E\left[\frac{\partial \log P_{\eta_{0}}\left(X_{t}|X_{t-1}\right)}{\partial \eta }\frac{\partial \log P_{\eta_{0}}\left(X_{t}|X_{t-1}\right)}{\partial \eta^{⊤}}\right]$ and $J\left(\eta_{0}\right)=E\left(\frac{\partial^{2}\ell \left(\eta_{0}\right)}{\partial \eta \partial \eta^{⊤}}\right)$.*


**Proof.** To prove the consistence of ${\hat{\eta }}^{cml}$, we denote $\ell_{t}\left(\eta \right)=\log P_{\eta }\left(X_{t}|X_{t-1}\right)$. Hence, $\ell \left(\eta \right)=\sum_{t=1}^{T}\ell_{t}\left(\eta \right)$. Similar to the first item of Theorem 4 in Chen et al. [18], we can verify that the assumptions of Theorem 4.1.2 in Amemiya [19] hold under Assumption 1, i.e., $E\ell_{t}\left(\eta \right)$ attains a strict local maximum at $\eta_{0}$; therefore, there exists an estimator ${\hat{\eta }}^{cml}$ such that ${\hat{\eta }}^{cml}\;\overset{a.s.}{\to }\;\eta_{0}$.In the following, we prove the asymptotic normality of ${\hat{\eta }}^{cml}$. It is easy to see $\partial \ell_{t}\left(\eta \right)/\partial \theta_{1}$, $\partial \ell_{t}\left(\eta \right)/\partial \theta_{2}$, and $\partial \ell_{t}\left(\eta \right)/\partial \nu$ exist and are three times continuous differentiable in Θ. Thus, there exist a $N(\eta_{0})$ such that $\partial^{2}\ell_{t}\left(\eta \right)/\left(\partial \eta \partial \eta^{⊤}\right)$ attains the maximum value at $\tilde{\eta }\in N\left(\eta_{0}\right).$ Therefore,
$$E∥\underset{\eta \in N(\eta_{0})}{\sup}\frac{\partial^{2}\ell_{t}\left(\eta \right)}{\partial \eta \partial \eta^{⊤}}∥=E∥\frac{\partial^{2}\ell_{t}\left(\tilde{\eta }\right)}{\partial \eta_{i}\partial \eta_{j}}∥<\infty .$$
Similar to the second item of Theorem 4 in [18], we can prove that
$$T^{-1}\sum_{t=1}^{T}\frac{\partial^{2}\ell_{t}\left(\eta \right)}{\partial \eta \partial \eta^{⊤}}\;\overset{p}{\to }\;E\left(\frac{\partial^{2}\ell_{t}\left(\eta_{0}\right)}{\partial \eta \partial \eta^{⊤}}\right)$$
by Theorem 4.1.3 in Amemiya [19]. Furthermore,
$$T^{-1}\sum_{t=1}^{T}\partial \ell_{t}\left(\eta_{0}\right)/\partial \eta \;\overset{p}{\to }\;E\left(\partial \ell_{t}\left(\eta_{0}\right)/\partial \eta \right)$$
by using ergodic theorem. Using the Martingale central limit theorem and the Cramér device, it is direct to show that
$$T^{-1/2}\partial \ell (\eta_{0})/\partial \eta \overset{d}{⟶}N\left(0,I\left(\eta_{0}\right)\right).$$
Then the asymptotic normal distribution of ${\hat{\eta }}^{cml}$ is obtained based on the Taylor series expansion of $\partial \ell \left({\hat{\eta }}^{cml}\right)/\partial \eta$ around $\eta_{0}$. □

## 4. Simulation

In this section, we conduct a simulation study to illustrate the large sample property of the CMPBAR(1) model.

In the simulation, we fix n=10, let sample size T=100,300,500, and use the optim function in R to optimize ℓ(η) in (Equation 10). To check the finite sample performance, we use the following parameter combinations of $\left(\theta_{1},\theta_{2},\nu \right)$ as
$$\begin{array}{cc} \; & (A1)=(0.25,0.25,0.5),\;(A2)=(0.25,1,0.5),\;(A3)=(0.25,1.5,0.5),(A4)=(1,1.5,0.5),\\ \; & (B1)=(0.25,0.25,1),\;(B2)=(0.25,1,1),\;(B3)=(0.25,1.5,1),(B4)=(1,1.5,1),\\ \; & (C1)=(0.25,0.25,1.5),\;(C2)=(0.25,1,1.5),\;(C3)=(0.25,1.5,1.5),(C4)=(1,1.5,1.5), \end{array}$$
where ν=0.5,1 and 1.5 to reflect overdispersion, equidispersion, and underdispersion, respectively.

For the simulated sample, performances of mean and standard deviation (sd) are given. For a scale parameter φ, $\mathrm{sd}=\sqrt{\frac{1}{m-1}\sum_{i=1}^{m}{\left({\hat{\varphi }}_{i}-\varphi \right)}^{2}}$, where ${\hat{\varphi }}_{i}$ is the estimator of φ in the *i*th replication and m=10,000. Summaries of the simulation results are given in Table 1, Table 2 and Table 3.

To illustrate the consistency and the asymptotic normality of the CML estimators, we present the boxplots of the CML estimates for (A1), (B1), and (C1) in Figure 4, Figure 5, and Figure 6, and their qqplots with T=500 in Figure 7, Figure 8, and Figure 9, respectively. Others are similar and we omit them.

These studies indicate that the CML method seems to perform reasonably well. First, Table 1, Table 2 and Table 3 show that the standard deviation of the CML estimator is decreasing with the sample size increase and the mean of the CML estimator is closer to the true parameter value in general cases. Second, Figure 4, Figure 5 and Figure 6 account for the location and dispersion of the estimates, all of which indicate the consistency of the estimators. Third, Figure 7, Figure 8 and Figure 9 indicate the asymptotic normality of the CML estimator.

## 5. Real Data Example

In this section, we consider the number of weekly rainy days for the period from 1 January 2005 to 31 December 2010 at Hamburg–Neuwiedenthal in Germany, where a week is defined as being from Saturday to Friday and n=7. The data were collected from the German Weather Service (http://www.dwd.de/, accessed on 12 December 2018). The sample path and the ACF and PACF plots of the observations are given in Figure 10 and Figure 11, respectively.

By computation, the sample mean and variance are 3.8371 and 3.6753, and the BID of the data is 1.2371, which implies the data exhibits extra-binomial variation. Hence, we use the CMPBAR(1) model, BAR(1) model [2], BBAR(1) model [9], and GBAR(1) model [11] to fit data by the CML method. We compare the estimated standard error (SE), −log-likelihood (−log-lik), Akaike’s information criterion (AIC) and Bayesian information criterion (BIC), which are summarized in Table 4, including the fitted results of the CML estimate.

From Table 4, the CMPBAR(1) model takes the smallest values of the −log-lik, AIC, and BIC. Hence, the CMPBAR(1) model might be more appropriate for the weekly rainy days.

To illustrate the adequacy of the CMPBAR(1) model, we consider the fitted Pearson residual analysis of the CMPBAR(1) model. By computation, the mean and variance of the fitted Pearson residual are 0.0760 and 1.0500, respectively. The residual analysis in Figure 12 shows that this model performs rather well.

In addition, to further check the adequacy of the CMPBAR(1) model, we present the probability integral transform (PIT) (if the fitted model is adequate, its PIT histogram looks like that of a uniform distribution, see [10] for more discussion) in Figure 13 based on the fitted CMPBAR(1) model.

As can be seen in Figure 13, the PIT histogram of the CMPBAR(1) model is close to uniformity, i.e., the PIT histogram confirms that the fitted CMPBAR(1) model works reasonably well for the weekly rainy days.

## 6. Concluding Remarks

This paper considers a new CMPB thinning operator and proposes a new CMPBAR(1) model, which provides an available method to model bounded data with under-dispersion, equi-dispersion, and over-dispersion. We discuss some properties of the new model, the estimate of the parameters, and its large-sample properties. Simulations are conducted to examine the finite sample performance of estimators. A real data example is provided to illustrate the applicability of the CMPBAR(1) model.

There are several directions in which we plan to take this work forward. First, the random coefficient CMPBAR(1) model can be introduced by
$$X_{t}=\alpha_{t}\circ_{\nu }X_{t-1}+\beta_{t}\circ_{\nu }\left(n-X_{t-1}\right),$$
where $\alpha_{t}=\alpha \left(X_{t-1}/n\right)$ and $\beta_{t}=\beta \left(X_{t-1}/n\right)$, “$\circ_{\nu }$” is the CMPB thinning operator and the counting series in “$\alpha_{t}\circ_{\nu }$”, and that in “$\beta_{t}\circ_{\nu }$” is independent and all of the counting series at time *t* is independent of $\left\{X_{s},s<t\right\}$; see Weiß and Pollett [8] for the random coefficient BAR(1) model. Second, a correlated sign-thinning operator can be established by
$$\alpha {⋄}_{\nu }X=\mathrm{sign}\left(\alpha \right)\mathrm{sign}\left(X\right)\sum_{i=1}^{X}Z_{i},$$
where sign(*x*) = 1 if x≥0 and sign(*x*)=−1 if x<0, $\left\{Z_{i},i=1,2,\cdots ,X\right\}$ is an exchangeable Bernoulli variable sequence with its pmf taking the form (Equation 5). Based on the correlated sign thinning operator, one can construct a Z-valued autoregressive model to analyze data with a range Z and under-dispersed, equi-dispersed, and over-dispersed. Third, a class of Conway–Maxwell–Poisson-binomial generalized autoregressive conditional heteroskedasticity models can be considered by
$$\begin{array}{c} Z_{t}|F_{t-1}∼\mathrm{CMPB}\left(n,\alpha_{t},\nu \right),\;\alpha_{t}=g_{\eta }\left(Z_{t-1}/n,\alpha_{t-1}\right), \end{array}$$
where η is the parameter vector involving in the model (see Ristić et al. [20] and Chen et al. [18] for ARCH-type models, Lee and Lee [21] and Chen et al. [22] for GARCH-type models for bounded data). In addition, a semi-parameter version can be considered by
$$\begin{array}{c} Z_{t}|F_{t-1}∼\mathrm{CMPB}\left(n,\alpha_{t},\nu \right),\;\alpha_{t}=g_{\eta }\left(Z_{t-1}/n,\alpha_{t-1}\right)+f_{\gamma }\left(X_{t}\right), \end{array}$$
where η is the parameter vector involved in the model, $\left\{X_{t}\right\}$ is the covariate process imposed in the observe process $\left\{Z_{t}\right\}$, and γ is the parameter vector involving in f(·).

## Figures and Tables

**Figure 1 entropy-25-00126-f001:**
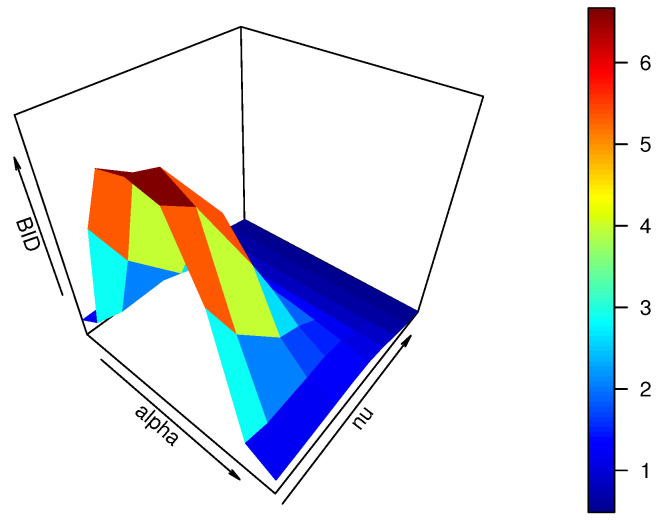
Plot of the BID of the CMPB distribution for different choices of α and ν.

**Figure 2 entropy-25-00126-f002:**
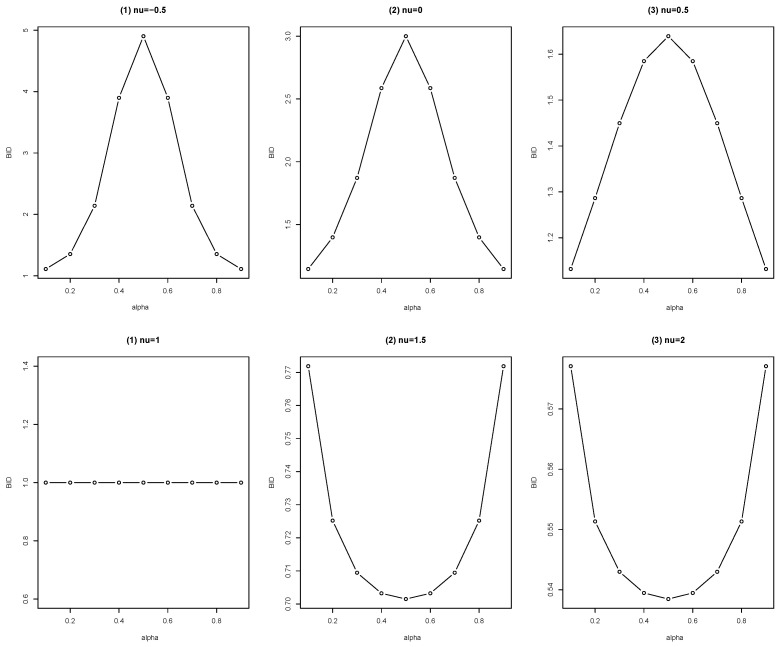
Plots of the BID of the CMPB distribution for different choices of α.

**Figure 3 entropy-25-00126-f003:**
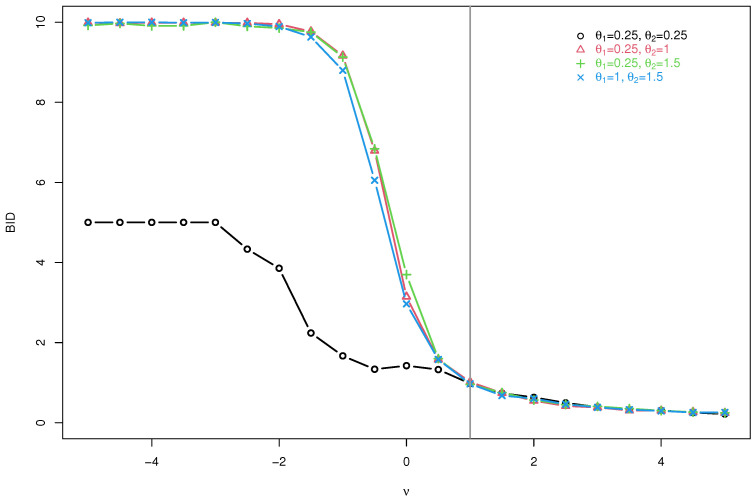
Plots of BID of the CMPBAR model.

**Figure 4 entropy-25-00126-f004:**
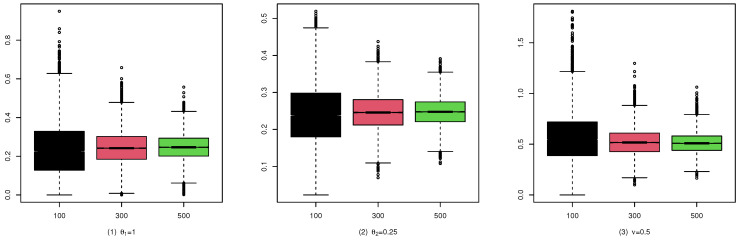
Boxplots of the CML estimates for (A1).

**Figure 5 entropy-25-00126-f005:**
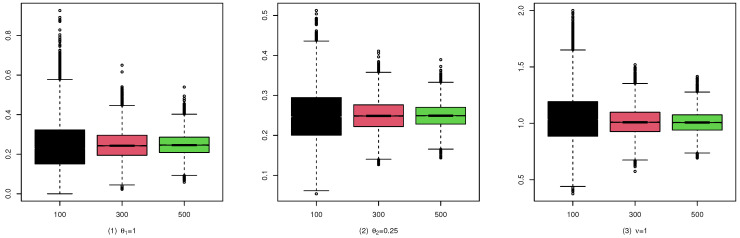
Boxplots of the CML estimates for (B1).

**Figure 6 entropy-25-00126-f006:**
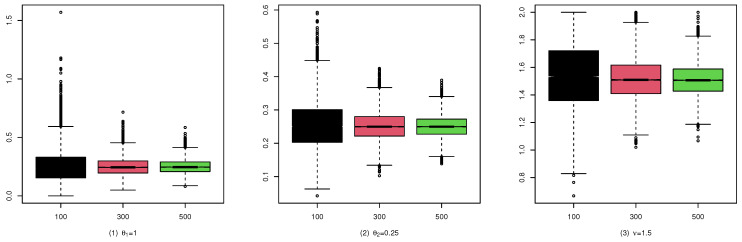
Boxplots of the CML estimates for (C1).

**Figure 7 entropy-25-00126-f007:**
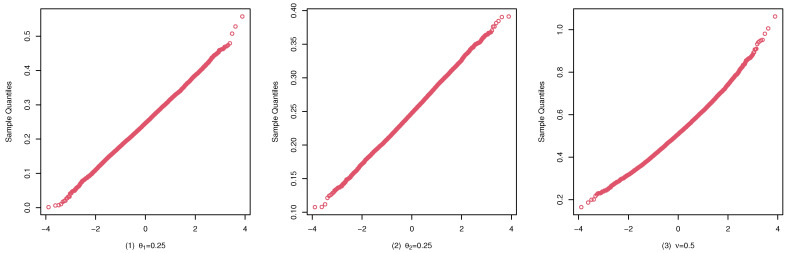
qqplots of the CML estimates for (A1) with T=500.

**Figure 8 entropy-25-00126-f008:**
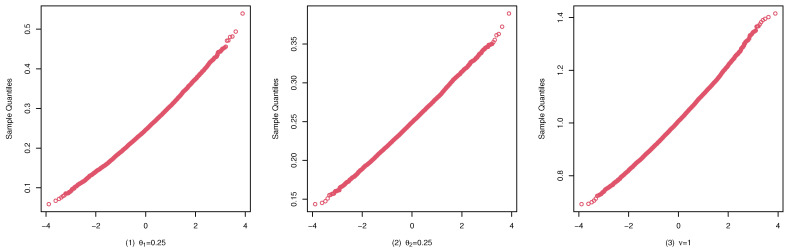
qqplots of the CML estimates for (B1) with T=500.

**Figure 9 entropy-25-00126-f009:**
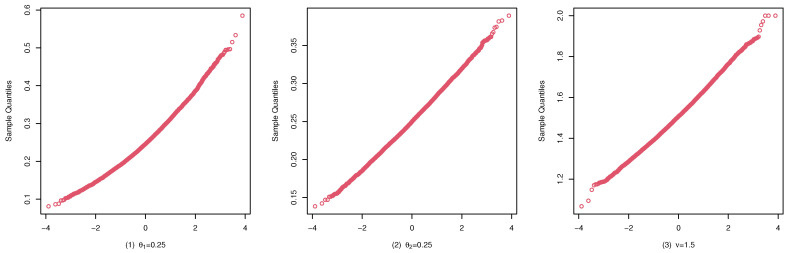
qqplots of the CML estimates for (C1) with T=500.

**Figure 10 entropy-25-00126-f010:**
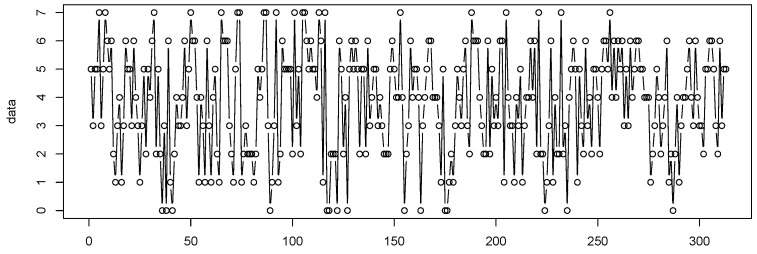
Path of the weekly rainy days.

**Figure 11 entropy-25-00126-f011:**
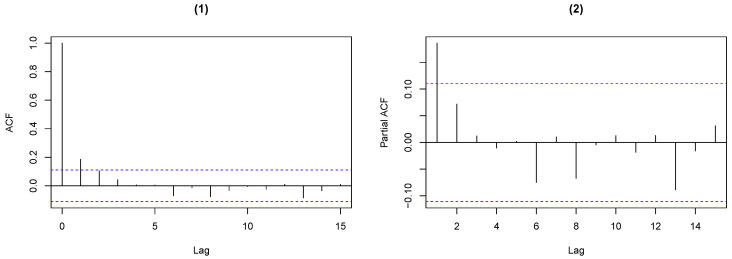
ACF and PACF plots of the weekly rainy days. (**1**) shows that the ACF exhibits significant value for lag 1, and (**2**) presents that the PACF indicates an AR(1)-like autocorrelation structure.

**Figure 12 entropy-25-00126-f012:**
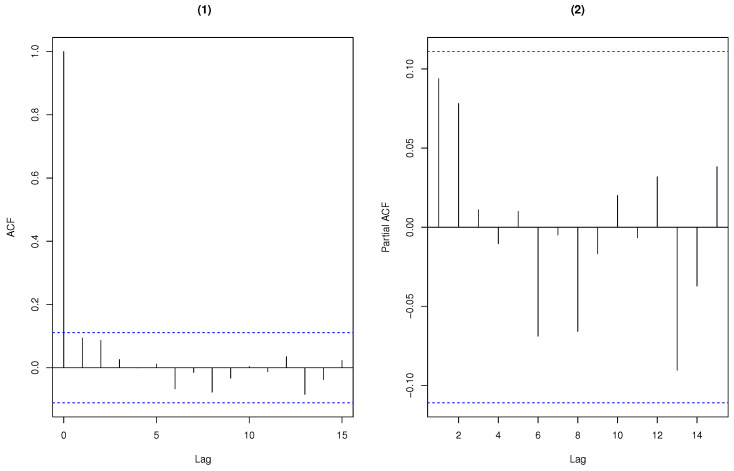
Pearson residual analysis of the weekly rainy days. (**1**) ACF (**2**) PACF.

**Figure 13 entropy-25-00126-f013:**
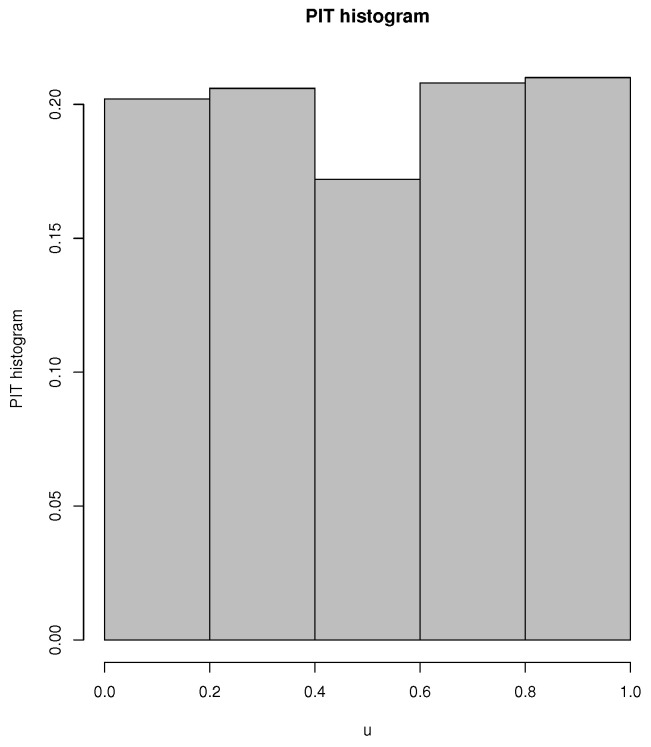
PIT histogram based on the fitted CMPBAR(1) model.

**Table 1 entropy-25-00126-t001:** Mean and sd in parentheses of estimates for (A1)–(A4).

	100	300	500
	(A1) = (0.25, 0.25, 0.5)		
$\theta_{1}$	0.2336 (0.1425)	0.2435 (0.0881)	0.2471 (0.0683)
$\theta_{2}$	0.2408 (0.0829)	0.2467 (0.0498)	0.2479 (0.0391)
ν	0.5682 (0.2484)	0.5231 (0.1371)	0.5135 (0.1065)
	(A2) = (0.25, 1, 0.5)		
$\theta_{1}$	0.2420 (0.0847)	0.2471 (0.0477)	0.2483 (0.0369)
$\theta_{2}$	1.0058 (0.0935)	1.0022 (0.0510)	1.0010 (0.0390)
ν	0.5236 (0.1353)	0.5070 (0.0742)	0.5044 (0.0567)
	(A3) = (0.25, 1.5, 0.5)		
$\theta_{1}$	0.2450 (0.0644)	0.2483 (0.0374)	0.2490 (0.0288)
$\theta_{2}$	1.5283 (0.1677)	1.5092 (0.0936)	1.5053 (0.0710)
ν	0.5269 (0.1505)	0.5072 (0.0821)	0.5046 (0.0628)
	(A4) = (1, 1.5, 0.5)		
$\theta_{1}$	1.0032 (0.1132)	1.0002 (0.0622)	1.0005 (0.0481)
$\theta_{2}$	1.5446 (0.2335)	1.5176 (0.1389)	1.5097 (0.1066)
ν	0.5246 (0.1336)	0.5087 (0.0755)	0.5052 (0.0585)

**Table 2 entropy-25-00126-t002:** Mean and sd in parentheses of estimates for (B1)–(B4).

	100	200	500
	(B1) = (0.25, 0.25, 1)		
$\theta_{1}$	0.2442 (0.1286)	0.2475 (0.0755)	0.2487 (0.0586)
$\theta_{2}$	0.2484 (0.0693)	0.2497 (0.0402)	0.2496 (0.0313)
ν	1.0484 (0.2317)	1.0152 (0.1288)	1.0094 (0.0997)
	(B2) = (0.25, 1, 1)		
$\theta_{1}$	0.2483 (0.0906)	0.2491 (0.0508)	0.2496 (0.0393)
$\theta_{2}$	1.0114 (0.1667)	1.0033 (0.0906)	1.0016 (0.0692)
ν	1.0390 (0.2070)	1.0130 (0.1140)	1.0083 (0.0873)
	(B3) = (0.25, 1.5, 1)		
$\theta_{1}$	0.2507 (0.0770)	0.2497 (0.0440)	0.2499 (0.0339)
$\theta_{2}$	1.5215 (0.2412)	1.5097 (0.1417)	1.5053 (0.1082)
ν	1.0409 (0.2167)	1.0128 (0.1201)	1.0084 (0.0922)
	(B4) = (1, 1.5, 1)		
$\theta_{1}$	1.0219 (0.1985)	1.0042 (0.1127)	1.0028 (0.0876)
$\theta_{2}$	1.5420 (0.3113)	1.5251 (0.2070)	1.5151 (0.1632)
ν	1.0318 (0.1883)	1.0114 (0.1057)	1.0067 (0.0820)

**Table 3 entropy-25-00126-t003:** Mean and sd in parentheses of estimates for (C1)–(C4).

	100	200	500
	(C1) = (0.25, 0.25, 1.5)		
$\theta_{1}$	0.2563 (0.1402)	0.2517 (0.0784)	0.2514 (0.0611)
$\theta_{2}$	0.2550 (0.0732)	0.2513 (0.0435)	0.2506 (0.0336)
ν	1.5431 (0.2529)	1.5169 (0.1553)	1.5103 (0.1191)
	(C2) = (0.25, 1, 1.5)		
$\theta_{1}$	0.2586 (0.1141)	0.2524 (0.0620)	0.2515 (0.0479)
$\theta_{2}$	1.0332 (0.2637)	1.0094 (0.1449)	1.0052 (0.1120)
ν	1.5408 (0.2482)	1.5157 (0.1497)	1.5100 (0.1153)
	(C3) = (0.25, 1.5, 1.5)		
$\theta_{1}$	0.2625 (0.1000)	0.2523 (0.0559)	0.2515 (0.0433)
$\theta_{2}$	1.5186 (0.3340)	1.5169 (0.2200)	1.5100 (0.1730)
ν	1.5383 (0.2512)	1.5167 (0.1531)	1.5103 (0.1180)
	(C4) = (1, 1.5, 1.5)		
$\theta_{1}$	1.0528 (0.2914)	1.0134 (0.1701)	1.0075 (0.1329)
$\theta_{2}$	1.5339 (0.3820)	1.5310 (0.2724)	1.5221 (0.2243)
ν	1.5398 (0.2350)	1.5161 (0.1396)	1.5100 (0.1082)

**Table 4 entropy-25-00126-t004:** Estimates for the weekly rainy days and SE are shown in parentheses.

Model	Estimates			−log-lik	AIC	BIC
	$\hat{\pi }$	$\hat{\rho }$				
BAR(1)	0.5476	0.1323		691.5400	1387.0800	1394.5720
	(0.0122)	(0.0325)				
	$\hat{\pi }$	$\hat{\rho }$	$\hat{\phi }$			
BBAR(1)	0.5475	0.1408	0.2827	623.6617	1253.33233	1264.5619
	(0.0177)	(0.0507)	(0.0320)			
	$\hat{\pi }$	$\hat{\rho }$	$\hat{\phi }$			
GBAR(1)	0.5493	0.1396	0.5209	625.4958	1256.9916	1268.2303
	(0.0169)	(0.0492)	(0.0279)			
	${\hat{\theta }}_{1}$	${\hat{\theta }}_{2}$	$\hat{\nu }$			
CMPBAR(1)	1.2313	0.9547	0.0995	622.6669	1251.3337	1262.5723
	(0.0627)	(0.0532)	(0.0681)			

## Data Availability

The weekly rainy days for the period from 1st January 2005 to 31st December 2010 at Hamburg–Neuwiedenthal in Germany is collected from the German Weather Service (http://www.dwd.de/ accessed on 12 December 2018), where a week is defined as being from Saturday to Friday and n=7 and the data can be found in Appendix A.

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
