# Peer review of "A Conway–Maxwell–Poisson-Binomial AR(1) Model for Bounded Time Series Data"

_entropy, 2023, doi:10.3390/e25010126_

Round 1

Reviewer 1 Report (Previous Reviewer 2)

Thanks for including the proofs

Author Response

Thank you very much for your helpful suggestions,
which are all valuable for revising and improving our paper.
In the revision, we give a point-to-point reply  and highlight the main changed places by red color.

Reviewer 2 Report (New Reviewer)

See attached file

Author Response

Thank you very much for your helpful suggestions, which are all valuable for revising and improving our paper. Below is our point-to-point reply. Your comments are in italic.\bigskip

{\it 1. Section 2 : Model formulation and stability properties\\
I would request the Authors describe the content in detail, without omission. Eventually, this section is the main content of the article.
\begin{itemize}
\item
On p3, there is no explanation for ``exchangeability or order 2";
\item
On p3, especially, the formula in (4) is confused, because of the both use of $k$ and $z$. I think the explicit formula (14) in Shmueli (2005) is more clear.
\item
On p3, line 68, the Authors derived the formula, however, the formula have never been used anyplace in this article.
\item On p3, the last line, ``Bin$(m, \alpha)$" should be ``Bin$(n, \alpha)$".
\item On p4, line 72, ``we let $\theta = \lambda^\nu$". What is the $\lambda$?
\end{itemize}}

{\bf Answer:} Thanks for your detailed and helpful suggestion.
Now, we add an explanation of the ``exchangeability or order 2", modify the pmf of $(Z_1,\cdots,Z_n)$ (i.e., the formula in (4)),
remove the $P(Z_i=0,Z_j=0)$, $P(Z_i=1,Z_j=0)$ and $P(Z_i=1,Z_j=1)$, $\forall i\neq j$,
correct $m$ as $n$, see page 4.

We consider a parameter transformation ($\theta = \lambda^\nu, \lambda>0$)
 to obtain $P(\alpha\diamond_{\nu}  n=i+1)=\left(\frac{n-i-1}{n-i}\right)
P(\alpha\diamond_{\nu}  n=i),$ which is necessary to  generate the random number of ``$\alpha\diamond_{\nu}X$''.\bigskip

{\it 2. Section 4: Simulation Study There are 5 scenarios in this section, (A1)-(A5), and (A1)- (A4) have $\nu$=1, which is usual binomial distribution and (A5) has $\nu$=2 corresponds to underdispersion. However, the results shown in this article are corresponding results of (A1). It still does not make sense to me. If the main characteristics of CMPBAR(1) model is allowable to analyze bounded data with overdispersion and underdispersoon, then why do not the Authors consider both two cases in the simulation design? To be frank with you, I would think that section 4 is rewritten.}

{\bf Answer:}This is a good question. We are sorry that we have not given the simulation study for $\nu<1$. Now, we choose $(\theta_1,\theta_2,\nu)$ as
\begin{align*}
&({\rm A1})=(0.25,0.25,0.5),~({\rm A2})=(0.25,1,0.5),
~({\rm A3})=(0.25,1.5,0.5),({\rm A4})=(1,1.5,0.5);\\
&({\rm B1})=(0.25,0.25,1),~({\rm B2})=(0.25,1,1),
~({\rm B3})=(0.25,1.5,1),({\rm B4})=(1,1.5,1);\\
&({\rm C1})=(0.25,0.25,1.5),~({\rm C2})=(0.25,1,1.5),
~({\rm C3})=(0.25,1.5,1.5),({\rm C4})=(1,1.5,1.5);
\end{align*}
and reconduct the simulation. The simulation results are given in Tables 1-3.
To illustrate the consistency and the asymptotic normality of the  CML estimators,
we present the boxplots and qqplots of the  CML estimates for (A1), (B1) and (C1) in Figures 4-5, 6-7 and 8-9. Others are similar and we omit them. See pages 7-10.

Reviewer 3 Report (New Reviewer)

I am enclosing all comments as an attached pdf file for the Authors and Editors.

Author Response

Thank you very much for your helpful suggestions, which are all valuable for revising and improving our paper. Below is our point-to-point reply. Your comments are in italic.\bigskip

{\it 1. Abstract, line 2: delete ``which is the number of the occurrence in n independent units" from the text.}

{\bf Answer:} Thanks for your detailed and helpful suggestion. Now, we  remove it, see page 1.\bigskip

{\it 2. Introduction: delete ``and are often under-dispersed ... over-dispersed (or extra-binomial variation)." from the text.}

{\bf Answer:} Thanks for your detailed and helpful suggestion.
Now, we  remove it, see page 1.\bigskip

{\it 3. The definition of the binomial thinning operator in (1) is incomplete. Furthermore, in (1) ``="should be rewritten as ``:=".}

{\bf Answer:} Thanks for your detailed and helpful suggestion. Now, we  correct it, see page 1.\bigskip

{\it 4. page 2, lines 20-21: delete ``See Wei{\ss}  and Testik [3] and Chen et al. [4] for the BAR model with outliers." from the text.}

{\bf Answer:} Thanks for your suggestion. Now, we  remove it, see page 1.\bigskip

{\it 5. page 2, line 26: the operator $\alpha\circ_{\phi}$ is introduced but not defined.}

{\bf Answer:} Thanks for your  suggestion. Now, we  correct it, see page 1.\bigskip

{\it 6. page 2, lines 26-28: the sentence ``It is worth mentioning that both the BAR(1) and the BBAR(1) model are aimed at a system with $n$ independent units, but not a system with $n$ dependent units." is not understandable as it is. Please, elaborate.}

{\bf Answer:} Thanks for your detailed and helpful suggestion. Now, we rephrase it, see page 2.

{\it 7.page 2, line 29: the operator $\alpha\circ_{\theta}$  is introduced but not defined.}

{\bf Answer:} Thanks for suggestion. Now, we  correct it, see page 1.\bigskip

{\it 8. page 2, lines 30-31: the sentence ``See [7], [8], [9] and [10] for the heteroscedastic models of the time series with finite range." should be removed from the text.}

{\bf Answer:} Thanks for your suggestion. Now, we  remove it, see page 1.\bigskip

{\it 9. page 3: what do we learn from Figure 1?}

{\bf Answer:} Figure 1 gives the pmf of the CMPB distribution for different choices of $\theta$ and $\nu$. To avoid  unnecessary confusion, now, we remove it.\bigskip

{\it 10.page 3, line 69: the expressions for $P(Z_i = 0, Z_j = 0), P(Z_i = 1, Z_j = 0)$, and $P(Z_i = 1, Z_j = 1)$ should be removed from the text.}

{\bf Answer:} Thanks for your suggestion. Now, we add the discussion with regard to exchangeability with order 2 and  remove these expressions, see page 4. \bigskip

{\it 11. page 3: in definition 1, replace ``=" by ``:=". Furthermore, the authors should be clearly explain the concept ``exchangeability of order 2"}

{\bf Answer:} Thanks for your suggestion. Now, we  correct it and add the discussion with regard to exchangeability with order 2, see page 4.\bigskip

{\it 12. page 4: the expression $M_{\alpha\diamond_{\nu}  X}(s)$ does not represent the mgf of $\alpha\diamond_{\nu}  X$}

{\bf Answer:} You are right that $M_{\alpha\diamond_{\nu}  X}(s)=E(e^(s\alpha\diamond_{\nu}  X))$. We are sorry that we wrote it as $E(e^(s\alpha\diamond_{\nu}  X)|X=n)$. Now, we correct it.

By Definition 1, $\alpha\diamond_{\nu}  X|(X=n) \sim {\rm CMPB}(n,\alpha,\nu)$. Hence, the mgf of $\alpha\diamond_{\nu}  X|(X=n)$ is that of the ${\rm CMPB}(n,\alpha,\nu)$. After a trade-off, we give the mgf of ${\rm CMPB}(n,\alpha,\nu)$ in the brief of the CMPB distribution, see page 3.

To account for the dynamic change of mean, variance BID, when $\alpha$ and $\nu$ are varying form $\{0.1,0.2,0.3,\cdots,0.9\}$ and $\{-2,-1.5,-0.5,0,0.5,1,1.5,2,2.5\}$, we give an example in Figure 1 with $n=7$.
Furthermore, for given $\nu$ (taking values in $\{-0.5,0,0.5,1,1.5,2\}$), Figures 2 and 3 give the plots of BID in  when $\alpha$ are varying form $\{0.1,0.2,0.3,\cdots,0.9\}$. See pages 3-4. \bigskip

{\it 13. page 4: the expressions $S^{'}(\theta,\nu)$ and $S^{''}(\theta,\nu)$ are introduced but not defined.}

{\bf Answer:} Thanks for your suggestion. Thanks for your suggestion.
Now, we  rephrase it, see page 3.

Round 2

Reviewer 2 Report (New Reviewer)

In the revised paper the authors made their revision satisfying the major comments addressed. In my opinion the quality of the paper is much improved.

I recommend acceptance.

Author Response

Thank you for your comments.

Reviewer 3 Report (New Reviewer)

The Authors modified  all details which I pointed out in the first draft,

so this revised draft is sufficient to be published in Entropy.

Author Response

Thank you for your comments.

This manuscript is a resubmission of an earlier submission. The following is a list of the peer review reports and author responses from that submission.

Round 1

Reviewer 1 Report

There are some things in the presentation of the model that can be improved.

1. The authors can provide the definition of the operator "\circle" (eqn. 1).

2. The definition 1 of the operator \diamond_\nu is unclear. In fact, it looks like a theorem, as it states, that "if  \diamond_\nu X = \sum_{i=1}^n Z_i, then  \diamond_\nu X | X = n \sim CMPB(n, \theta, \nu)". It makes an impresion that it is a statement about conditional expectation of some random variable, and not a definition. Moreover, "n \sim CMPB(n, \theta, \nu)" suggest that a constant (n) is a random variable with CMPB distribution.

3. The authors could provide more information about the intuition behind proposed model. In particular they could write, why they consider it as a good tool for modeling numbers of rainy days.

Author Response

Reply to Reviewer #1

Special thanks for your helpful suggestions and comments. We have made a complete revision which we hope to meet with your approval. Below is our point-to-point reply. Your comments are in italic.

  1. The authors can provide the definition of the operator "\circle" (eqn. 1).

Answer: We would like to appreciate this detailed and helpful suggestion. Now, we add the definition of the thinning operator, see page 1.

  1. The definition 1 of the operator \diamond_\nu is unclear. In fact, it looks like a theorem, as it states, that "if  \diamond_\nu X = \sum_{i=1}^n Z_i,then  \diamond_\nu X | X = n \sim CMPB(n, \theta, \nu)". It makes an impresion that it is a statement about conditional expectation of some random variable, and not a definition. Moreover, "n \sim CMPB(n, \theta, \nu)" suggest that a constant (n) is a random variable with CMPB distribution.

Answer:These are good suggestions. Now, the Definition 1 is rephrased, see page 3. As for "n \sim CMPB(n, \theta, \nu)", we are sorry that we have not made it clear for the first time. Now, it is rephrased, see page 3.

  1. The authors could provide more information about the intuition behind proposed model. In particular they could write, why they consider it as a good tool for modeling numbers of rainy days.

Answer:Thanks for your detailed and helpful suggestion. Now, we re-explained the intuition (see page 2) from the perspective of the dispersion but not from the perspective of the real data, because the BIDs of the BBAR(1) model and the GBAR(1) model are greater than 1, i.e., both the BBAR(1) model and the GBAR(1) model are suitable to analyze bounded data with over-dispersion. Hence, we considered the rainy days counts with over-dispersion to illustrate the flexible and superior of the proposed model.

Reviewer 2 Report

Chen et al presents an interesting paper in which they introduce and study the properties of CMPB thinning operator and further use it to define CMPBAR(1) model as an extension of the classical BAR(1) model.

I think the content presented in this paper is novel and adds nicely to the existing time series literature.

I checked most of the calculations and they look correct to me. My only concern is that the proofs for Theorems 1, 2, and 3 are omitted by stating that the proofs are similar to proofs in other papers. These other referenced papers are not in open access journals to the best of my understanding, and hence may not be accessible in many regions of the world. Since Entropy is an open access journal and accessible to everyone, it will be helpful for the wider community if the proofs (even if similar) are presented in the current paper. At least a sketch of each proof should be presented.

There are minor typos that need to be fixed. e.g. 'examples' in line 33.

Author Response

Reply to Reviewer #2

Special thanks for your helpful suggestions and comments. We have made a complete revision which we hope to meet with your approval. Below is our point-to-point reply. Your comments are in italic.

  1. I checked most of the calculations and they look correct to me.My only concern is that the proofs for Theorems 1, 2, and 3 are omitted by stating that the proofs are similar to proofs in other papers. These other referenced papers are not in open access journals to the best of my understanding, and hence may not be accessible in many regions of the world. Since Entropy is an open access journal and accessible to everyone, it will be helpful for the wider community if the proofs (even if similar) are presented in the current paper. At least a sketch of each proof should be presented.

Answer: We would like to appreciate your detailed and helpful suggestion, we have added the proof of Theorems 1, 2 and 3, see pages 4, 6 and 7, respectively.

  1. There are minor typos that need to be fixed. e.g. 'examples' in line 33.

Answer:Thanks for your detailed and helpful suggestion, we have carefully checked the full text and corrected the problems.
